# Positive Changes in Body Composition and Profiles of Individuals with Diabetes 3 Years Following Laparoscopic Sleeve Gastrectomy in Japanese Patients with Obesity

**DOI:** 10.3390/nu16223926

**Published:** 2024-11-18

**Authors:** Yoshinori Ozeki, Takayuki Masaki, Shotaro Miyamoto, Yuichi Yoshida, Mitsuhiro Okamoto, Koro Gotoh, Yuichi Endo, Masafumi Inomata, Hirotaka Shibata

**Affiliations:** 1Department of Endocrinology, Metabolism, Rheumatology and Nephrology, Faculty of Medicine, Oita University, Yufu City 879-5593, Japan; ozeki23@oita-u.ac.jp (Y.O.); shoutarou1029@oita-u.ac.jp (S.M.); y-yoshida@oita-u.ac.jp (Y.Y.); mokamoto@oita-u.ac.jp (M.O.); gotokoro@oita-u.ac.jp (K.G.); 2Obesity and Diabetes Center for Advanced Medicine, Faculty of Medicine, Oita University, Yufu City 879-5593, Japan; endo@oita-u.ac.jp; 3Department of Practical Nursing Sciences, Faculty of Medicine, Oita University, Yufu City 879-5593, Japan; 4Faculty of Welfare and Health Sciences, Oita University, Oita City 870-1192, Japan; 5Department of Gastroenterological and Pediatric Surgery, Faculty of Medicine, Oita University, Yufu City 879-5593, Japan; inomata@oita-u.ac.jp

**Keywords:** body composition, laparoscopic sleeve gastrectomy, obesity, diabetes, fat mass

## Abstract

Background and Objectives: We analyzed the changes in obesity, glucose metabolism, and body composition over a 3-year period in Japanese patients with obesity following laparoscopic sleeve gastrectomy (LSG). Methods: Body weight, parameters related to diabetes such as glycated hemoglobin (HbA1c), and electrical impedance analysis were used to assess body composition in forty-eight Japanese patients with obesity before surgery and 6 months, 1 year, 2 years, and 3 years after LSG. Results: At 6 months, 1, 2, and 3 years post-LSG, there were significant reductions in body weight, body mass index, blood pressure, fasting plasma glucose, triglyceride, and HbA1c levels. Six months after LSG, fat mass (FM), muscle mass (MM), and %FM all showed a decrease compared to pre-treatment values (all *p* < 0.05). FM and %FM remained in a decreased state until 3 years had passed. In contrast, %MM increased at 6 months post-LSG and was maintained up to 3 years post-LSG (all *p* < 0.05). Furthermore, changes in FM and %FM were associated with changes in body weight and A1C. In contrast, change in %MM exhibited a negative correlation with body weight and A1C following LSG. Finally, multivariate regression analyses demonstrated that alterations in FM were independent factors affecting body weight in patients with obesity 3 years after LSG. Conclusions: We observed improvements in FM, fasting plasma glucose, and HbA1c levels over a 3-year period in Japanese patients after LSG. The reduction in FM and maintenance of %MM after LSG were suggested as possible links between the effects of LSG on obesity and diabetes over 3 years.

## 1. Introduction

Obesity and type-2 diabetes mellitus (T2DM) lead to several complications, such as cardiovascular and renal diseases, and are important issues worldwide, including Japan [1,2,3]. Appropriate drug therapy and bariatric surgery are important in the treatment of obesity and T2DM [4]. Among the obese surgical treatments, laparoscopic sleeve gastrectomy (LSG) has been performed extensively in European countries and the United States for the treatment of severe obesity and T2DM [5,6]. 

However, there are relatively few cases of LSG in Japan compared to those in European countries and the United States; therefore, there are limited studies reporting the long-term effects of LSG on body weight (BW), body composition, and profiles of individuals with diabetes in Japanese patients [7].

A method to assess the outcomes of bariatric surgery is by utilizing the body mass index (BMI) [8]. Nevertheless, weight loss and BMI by themselves are inadequate for evaluating the impact of LSG on obesity and T2DM. Increases in BMI and body composition, especially visceral fat mass (FM), are associated with an increased risk of T2DM [9]. Decrease of muscle mass (MM) and excessive fat accumulation are both associated with T2DM [10,11]. MM was analyzed in terms of total MM and skeletal MM (TMM and SMM), %MM, upper and lower extremity MM, and the ratio of lower limb MM to body weight (L/W) [12,13].

Bioelectrical impedance analysis (BIA) has proven to be an easy approach for evaluating body composition [14,15,16]. Recent studies have investigated the impact of LSG using BIA [17,18,19]. Nonetheless, the long-term impact of bariatric surgery on body composition in Japanese patients with obesity is still unclear.

Body composition varies greatly depending on race and the environment [20]. The proportion of body fat is different in Japanese people compared to European and American Caucasians [21,22,23]. Although some studies in Western countries have reported on the body composition of patients with obesity, long-term changes in glucose metabolism and body composition, such as MM and FM after LSG in Japanese patients, are unclear. Therefore, to investigate the long-term benefits of LSG in Japanese patients with obesity, we attempted to identify new findings based on changes in glucose homeostasis and body composition, including FM and MM, after LSG. This study examined long-term changes in glucose parameters and body composition assessed using BIA. We report the 3-year long-term changes in BW, body composition, and glucose metabolism after LSG.

## 2. Materials and Methods

### 2.1. Research Design and Participants

We enrolled 95 consecutive patients who performed LSG at Oita University Hospital between January 2014 and December 2021. Patients were treated according to the LSG surgical standards of the Japanese Ministry of Health, Labor, and Welfare. At 6 months and 1, 2, and 3 years after LSG, we examined the 3-year changes in glucose metabolism and body composition using BIA. Three years after LSG, changes in body composition and blood profiles were evaluated. There were 47 patients who lost follow-up due to the following reasons: transfer to another hospital (*n* = 23), COVID-19 infection (*n* = 5), and treatment interruption (*n* = 19) (Figure 1). Excluding patients who were unable to attend due to COVID-19, treatment interruption, or transfer to another hospital during this study, 48 patients (19 male and 29 females, mean body mass index [BMI] = 43.7 ± 8.7 kg/m^2^, mean age = 42.2 ± 9.3 years) ultimately participated in this study. Among 48 patients, 26 patients have T2DM, and 32 patients have hypertension. The present research complied with the Declaration of Helsinki and received approval from the Ethics Committee of Oita University.

### 2.2. Surgery

The surgical techniques for LSG procedures have been previously reported [24,25]. In summary, following visual access to the intraperitoneal cavity using a 10 mm Visiport trocar (US Surgical, Norwalk, CT, USA) placed 18 cm below the xiphoid process, the abdomen was inflated to 15 mmHg. Subsequently, a 15 mm port, a 12 mm port, and two 5 mm ports were positioned in the upper abdomen, and a Nathanson liver retractor (Automated Medical Products Corp, Edison, NJ, USA) was introduced through a 5 mm skin incision in the subxiphoid area to hold back the left lateral segment of the liver. The vessel sealing system (LigaSure system, Valleylab, Boulder, CO, USA) was used to dissect the greater omentum from 5 cm proximal to the pyloric ring to the angle of His. Following the placement of a 10.5 mm (32-Fr) endoscope with or without a 15 mm (45-Fr) endoscopy tube directly along the lesser curvature of the stomach, endoscopic linear staplers (45 or 60 mm in length, EndoGIA, US Surgical) were applied in a sequential manner, commencing 6 cm from the pylorus. Subsequently, the staple lines were reinforced to avoid bleeding and leakage. The specimen was obtained via the 15 mm port site. Ultimately, oral endoscopy was used to examine stenosis of the remnant stomach as well as hemorrhaging and air leakage from the staple line.

### 2.3. Collection of Measurement Parameters and Blood Data

Blood was collected between 8:00 and 11:00 h from the antecubital vein of subjects who had fasted overnight. Blood was collected from the participating patients and fasting plasma glucose (FPG), HbA1c, low-density lipoprotein (LDL), triglycerides, high-density lipoprotein (HDL), aspartate aminotransferase (AST), alanine aminotransferase (ALT), gamma GTP, blood urea nitrogen (BUN), and creatinine (Cr) levels. Blood samples were collected prior to LSG at the first consultation, 6 months, 1 year, 2 years, and 3 years after LSG.

### 2.4. Analysis of Body Weight and Body Composition

According to previous report, ΔBMI was calculated as (Initial BMI) − (Post-LSG BMI). Percent of total weight loss (%TWL) was determined by [(Initial Weight) − (Post-LSG Weight)]/[(Initial Weight)] × 100. Percent excess weight loss (%EWL) was calculated as [(Initial Weight) − (Post-LSG Weight)]/[(Initial Weight) − (Ideal Weight)] × 100 [26]. Body composition, including FM, total muscle mass (TMM), skeletal muscle mass (SMM), body fluid, and bone mineral content (BMC), was measured periodically using a BIA device (InBody 770; InBody Japan., Ltd., Tokyo, Japan) according to previous reports. TMM is also defined as MM in the device. Body FM, MM, percentage of fat mass (%FM), and MM (%MM) were calculated using the following formula: %FM was determined by multiplying the percentage of fat by body weight (kg). %MM was calculated as MM (kg)/body weight (kg) × 100. %SMM was determined by SMM (kg)/body weight (kg) × 100. Body composition data were collected before bariatric surgery and at 6 months, 1 year, 2 years, and 3 years following LSG.

### 2.5. Statistical Analyses

Continuous variables are represented as mean ± standard deviation (SD). We have used a Shapiro–Wilk and Levene’s test regarding the normality of the data distribution. Considering the normal distribution, analysis of variance (ANOVA) was utilized to evaluate the differences between baseline and postoperative data. The data at every time point were assessed using post-hoc multiple comparisons. The independent associations of these variables were evaluated using multiple regression analyses and simple correlation coefficients were examined. To evaluate the relationship between body composition and obesity, multiple regression analyses were performed, controlling potential confounders such as BMC. A *p*-value of less than 0.05 was considered statistically significant. All analyses were conducted using the JMP software (JMP14.1; SAS Institute, Cary, NC, USA).

## 3. Results

### 3.1. Clinical Characteristics of Patients and Changes in BW and BMI Following LSG

Table 1 displays the characteristics of the participants prior to surgery. Table 1 shows the changes in BW after LSG. None of the patients experienced serious adverse events or died during this study’s period. BW and BMI both decreased at 6 months and 1, 2, and 3 years after LSG compared with preoperatively (*p* < 0.01). There were no notable differences in BW and BMI after 6 months and at the 3 years. There were no differences in BW and BMI change between men and women after LSG (BW (men vs. women: *p* = 0.22) and BMI (men vs. women: *p* = 0.19).

### 3.2. Changes in Plasma Metabolic Parameters and Antidiabetic, Antihypertensive, and Lipid-Lowering Medications Use Following LSG

The glucose metabolic parameters FPG and HbA1c both decreased at 6 months and 1, 2, and 3 years after LSG compared to before treatment (*p* < 0.01, respectively) (Table 1). AST, ALT, GTP, and triglyceride levels decreased at 6 months and 1, 2, and 3 years after LSG compared to preoperative levels (*p* < 0.01). Conversely, plasma HDL-C levels increased at 6 months and 1, 2, and 3 years after LSG compared to preoperative levels (Table 1). There were no notable differences between 1 year and 2 years, 2 years vs. 3 years, or 1 year and 3 years for FPG, HbA1c, AST, ALT, GTP, LDL-C, and triglyceride levels (*p* > 0.1). There were no significant changes in BUN levels throughout this study’s period (*p* > 0.1). In the present study, 50% (24/48) of patients were taking antidiabetic medications (DPP-4 inhibitors, 12 patients; glucagon-like peptide 1 receptor agonists, 5 patients; SGLT2 inhibitors, 6 patients; metformin, 13 patients; sulfonylureas, 5 patients; insulin, 3 patients; and others, 7 patients) as type-2 diabetes prior to bariatric surgery. In addition, 67% (32/48) of patients were taking antihypertensive medications (angiotensin II receptor blockers, 18 patients; mineralocorticoid receptor antagonists, 1 patient; calcium channel blockers, 16 patients; and others, 7 patients). Additionally, 29% (14/48) of patients were taking lipid-lowering medications (Statins, 11 patients; Fibrates, 3 patients; and others, 1 patient). As shown in Table 1, the proportion of patients who were taking any glucose-lowering, anti-hypertension, or lipid-lowering medications all decreased 1, 2, and 3 years after surgery compared to the levels at pre-surgery, respectively (Table 1). The percentage of patients with diabetes remission was 62% (16/26) at 3 years after surgery. In addition, the percentage of patients with normal blood pressure without antihypertensive drugs was 56% (18/32).

### 3.3. Time-Course Changes of FM and, % FM After LSG

Table 2 shows changes in body composition following LSG surgery. Both FM and %FM were significantly reduced and maintained at 6 months, 1, 2, and 3 years compared to pre-LSG (*p* < 0.01, respectively) (Table 2). There were no notable differences in FM and %FM between years 1 and 2 and between years 2 and 3 (*p* > 0.1, respectively). There was no difference in FM change between men and women after LSG (*p* = 0.41).

### 3.4. Changes in Muscle Mass and the Ratio of Extra Cellular Fluid and Bone Mineral Content After LSG

TMM and SMM at 6 months, 1 year, 2 years, and 3 years decreased compared to the preoperative values (*p* < 0.01) (Table 2); however, TMM and SMM were maintained from 6 months to 3 years after LSG (*p* > 0.1). Compared with the pre-LSG values, both %TMM and %SMM increased at 6 months and 1, 2, and 3 years, respectively (Table 2). No significant differences were observed in MM or %MM between 1 and 2 years and between 2 and 3 years (*p* > 0.1). Body fluid was significantly reduced and maintained at 6 months and at 1, 2, and 3 years compared to pre-LSG (*p* < 0.01). Conversely, the ratio of extracellular fluid significantly increased at 6 months, 1 year, 2 years, and 3 years after LSG (*p* < 0.01). BMC was reduced at 2 and 3 years after LSG compared to preoperative levels. There were no differences in TMM and SMM changes between men and women after LSG (TMM (men vs. women: *p* = 0.27) and SMM (men vs. women: *p* = 0.25).

The upper and lower leg muscles both decreased at 6 months post-LSG (*p* < 0.01) and were maintained for up to 3 years post-LSG (Table 2). In contrast, the ratio of the lower leg muscles and body weight (L/W) and upper leg muscles and body weight (U/W) both increased at 6 months post-LSG (*p* < 0.01) (Table 2) and were maintained up to 3 years post-LSG. The ratio of the upper Skeletal MM to the lower Skeletal MM (U/L) significantly decreased at 6 months, 1 year, 2 years, and 3 years after LSG. This result showed that the lower MM was maintained more than the upper MM after LSG.

### 3.5. Relationship Between Alterations in Body Compositions and Variations in BW and Glycemic Metabolic Parameters 3 Years Post-LSG

Changes in BW were associated with changes in FM (r = 0.91; *p* < 0.01), %FM (r = 0.71; *p* < 0.01), TMM (r = 0.68; *p* < 0.01), %TMM (r = −0.67; *p* < 0.01), % SMM (r = −0.54; <0.01), U/W (r = −0.46; *p* < 0.01), and L/W (r = −0.55; *p* < 0.01). Changes in FPG levels were associated with changes in FM (r = 0.30; *p* = 0.05). Changes in HbA1c levels were correlated with changes in FM (r = 0.47; *p* < 0.01), %FM (r = 0.35; *p* = 0.02), TMM (r = 0.36; *p* = 0.02), and % total MM (r = −0.33; *p* = 0.03). Figure 2 shows the associations among delta-BW-FM, delta-FPG-FM, and delta-HbA1c-%TMM.

### 3.6. 3 Years Post-LSG, Multiple Regression Analyses Regarding Changes in Body Weight, FPG, and HbA1c

Multiple regression analyses were conducted to evaluate the relationship between body composition and glycemic parameters while controlling for possible confounders, such as BMC. Changes in BW were used as the dependent variable in multiple regression analyses, with FM, %FM, MM, %MM, U/W, and L/W as the independent variables. Alterations in FM were the factor independently linked to variations in BW (*p* < 0.01) (Table 3). Conversely, no individual factor showed a correlation with alterations in FPG or HbA1c levels (*p* > 0.05).

## 4. Discussion

This is the first research to assess prolonged alterations in weight loss, glucose/lipid metabolism, and body composition over a 3-year period after LSG in Japanese patients with obesity. Body weight, FPG, HbA1c, FM, and %FM decreased over 3 years. In contrast, % TMM and % SMM increased over the long period of 3 years after LSG. A continuous reduction of around 50% of excess body weight is regarded as successful in weight loss [26,27,28]. In this study, compared to the preoperative weight, the %EBWL was almost 60% after three years, which was largely maintained.

Our previous analysis of body composition 12-month after LSG in Japanese patients with obesity showed a significant decrease in FM [17,29]. This is significant considering that the loss of FM is important for the improvement of glucose and lipid metabolic disorders 12-month after surgery. In the present study, FM decreased even 3 years after LSG compared to preoperative levels. In addition, FPG and HbA1c levels decreased at 6 months and 1, 2, and 3 years after LSG compared with preoperative levels.

The loss of MM could be associated with a higher risk of developing diabetes. During the ongoing 3-year study, TMM at 6 months, 1 year, 2 years, and 3 years showed a decrease in comparison to the preoperative values, but TMM was maintained from 6 months to 3 years after LSG. In addition, for pre-LSG, both %TMM and %SMM increased at 6 months and 1, 2, and 3 years, respectively. Similarly, MM in the upper and lower extremities was maintained compared to the preoperative values. This finding regarding the maintenance of TMM and increase in % TMM may be related to the improvement in glucose metabolism even 3 years after surgery. As mentioned above, FPG and HbA1c levels both improved and were maintained over a long period of 3 years after LSG. Strengthening skeletal muscle benefits the glycemic profile through mechanisms such as improved glucose utilization, and enhanced muscle maintenance benefits glucose metabolism [13,30]. The results indicated that FM, %FM, and %MM were important risk factors for obesity and diabetes after 3 years of LSG. In the present study, univariate analysis showed that ΔBW and ΔFM were related, indicating an association between weight loss and fat mass reduction with LSG surgery. Reducing fat mass correlates with a decrease in fasting blood glucose levels. In contrast, %TMM and ΔHbA1c were inversely correlated, indicating that HbA1c levels increased as total muscle mass decreased. The weight loss induced by LSG was related to a decrease in fat mass, indicating that the enhancement in glucose tolerance parameters might be due to both a reduction in fat mass and an increase in muscle percentage. Because the MM is an energy-metabolizing active organ, a sustained significant decrease in the MM may lead to weight rebound via a decrease in the basal metabolic rate. Taken together, the decrease in FM, maintenance of MM, and increase in %MM may have contributed to the improved glucose metabolism. Finally, the multivariate analysis showed that weight loss was specifically defined by fat mass. In the analysis after 1 year of LSG, correlations between body fat and body weight, muscle mass, and A1C were found in the multivariate analysis [17]. In contrast, only a correlation between body fat and body weight was found three years after LSG. The current study found that a decrease in body fat was most closely related to weight loss.

In the present study, we discuss the mechanisms underlying decreased FM and MM maintenance. Possible mechanisms for the decrease in FM and MM maintenance are an increase in appropriate protein intake through nutritional guidance using body composition results in our institution and an overall increase in lower extremity physical leg activity due to weight loss. In fact, the lower MM is better maintained than the upper MM 3 years after LSG.

Many studies indicate that bariatric surgery leads to more sustained weight loss and reduced rebound rates compared to intensive medical therapy alone [31,32,33]. Surgery for obesity is more successful than traditional medical treatment in managing type-2 diabetes and enhancing life expectancy [29,31,34]. Nonetheless, there are limited studies indicating long-term changes in body composition, including body fat and muscle mass. It is important to examine the long-term changes in body composition after LSG. It has been shown that the body composition of Japanese subjects with obesity differs significantly from that of non-Japanese subjects. Japanese individuals with obesity are more likely to have abdominal fat and develop T2DM than Caucasian individuals [21,22]. It would also be interesting to investigate the association between variations in visceral adipose tissue and subcutaneous adipose tissue and impaired glucose metabolism in Japanese individuals with obesity following LSG.

The current study demonstrated that the impact of weight loss in Japanese individuals with obesity was maintained for an extended duration of three years. Loss of muscle mass and sarcopenia after LSG is an important issue in non-Japanese subjects [35]. In the present study, body composition indicated that a decrease in fat mass and preservation of % muscle mass could be sustained over a three-year period in Japanese subjects. It is a possibility that there are racial differences in MM maintenance.

## 5. Limitations

Our study had several limitations. Potential statistical overfitting due to small sample size and potential confounders. The first is the observational nature of the small sample size. The number of bariatric surgery cases per year in Japan is very small compared to other developed countries. Hence, large surgery cohorts involving sub-groups are very hard to perform in our institution. Second, this study could not identify the determining factors that influenced body composition, as lifestyle and therapy modifications may have influenced body composition after LSG. Third, the adiposity analysis did not separate subcutaneous and visceral fat in this study.

## 6. Conclusions

Bariatric surgery has a significant impact on body composition, including FM and MM. Over a 3-year period, we observed improvements in FM, fasting plasma glucose, and HbA1c levels among Japanese patients following LSG. The decrease in FM and the maintenance of %MM following LSG were proposed as potential connections between the impacts of LSG on obesity and diabetes over a period of 3 years. Well-designed studies are required to enhance the strategies for decreasing FM and preserving MM following LSG.

## Figures and Tables

**Figure 1 nutrients-16-03926-f001:**
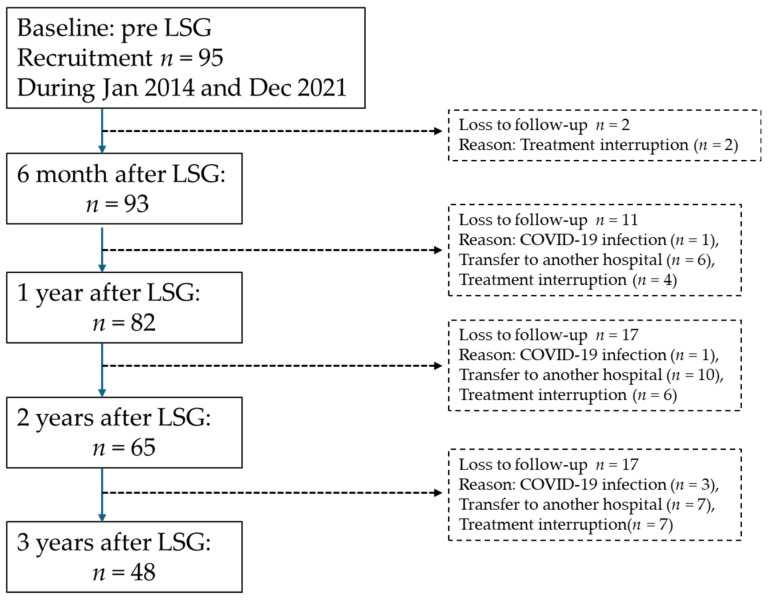
Flow chart of this study participants.

**Figure 2 nutrients-16-03926-f002:**
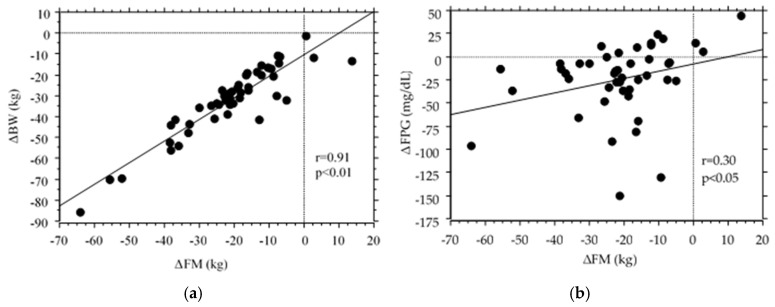
The relationship between variations in glycemic metabolic parameters and body composition. (**a**) Association between BW and FM, (**b**) association between FPG and FM, and (**c**) association between HbA1c and % TMM. Variables: Δ (0–3 years) variables, BW: body weight, MM: muscle mass, FPG: fasting plasma glucose, HbA1c: hemoglobin A1c. Correlation coefficients were simply calculated. r = correlation coefficient.

**Table 1 nutrients-16-03926-t001:** Basal clinical characteristics and time-course changes in BW, blood pressure, plasma parameters, and antidiabetic, antihypertensive, and lipid-lowering medications use.

	Pre-LSG	6 Months	1 Year	2 Years	3 Years
Body weight (kg)	116.1 ± 24.4	82.4 ± 17.2 **	80.0 ± 18.5 **	81.6 ± 22.0 **	84.6 ± 21.6 **
%TBWL		28.5 ± 8.3	30.7 ± 10.5	29.6 ± 12.7	27.2 ± 11.9
%EBWL		60.6 ± 19.5	64.8 ± 23.5	63.0 ± 27.7	57.3 ± 24.7
BMI (kg/m^2^)	43.7 ± 8.7	31.4 ± 6.7 **	30.2 ± 6.5 **	30.4 ± 7.4 **	31.7 ± 7.6 **
Systolic blood pressure (mmHg)	137.0 ± 17.5	119.9 ± 19.4 **	120.8 ± 16.0 **	125.1 ± 18.0 **	125.3 ± 16.6 **
Diastolic blood pressure (mmHg)	83.7 ± 12.4	73.0 ± 11.4 **	72.7 ± 11.1 **	75.6 ± 12.5 **	75.5 ± 12.5 **
Fasting plasma glucose (mg/dL)	116.5 ± 35.2	99.8 ± 26.7 **	96.8 ± 20.3 **	97.5 ± 26.9 **	92.9 ± 24.6 **
HbA1c (%)	6.8 ± 1.3	5.6 ± 0.8 **	5.7 ± 0.9 **	5.7 ± 0.9 **	5.7 ± 0.8 **
Triglycerides (mg/dL)	170.3 ± 84.8	103.4 ± 47.5 **	100.3 ± 77.8 **	97.4 ± 49.2 **	100.1 ± 55.2 **
HDL cholesterol (mg/dL)	48.1 ± 11.1	58.5 ± 16.2 **	63.7 ± 16.3 **	68.0 ± 19.0 **	70.6 ± 19.5 **
LDL cholestrol (mg/dL)	125.4 ± 32.3	121.5 ± 30.7	119.2 ± 32.1	112.6 ± 29.4	111.4 ± 29.0
BUN (mg/dL)	12.6 ± 3.8	13.2 ± 4.7	14.1 ± 4.3	13.6 ± 4.7	13.7 ± 4.1
Creatinine (mg/dL)	0.7 ± 0.2	0.7 ± 0.1 *	0.7 ± 0.2	0.7 ± 0.2 **	0.7 ± 0.2 **
AST (IU/L)	35.0 ± 26.1	17.1 ± 5.5 **	17.7 ± 4.5 **	17.7 ± 5.3 **	18.2 ± 4.5 **
ALT (IU/L)	49.6 ± 37.6	14.6 ± 5.8 **	16.5 ± 6.9 **	16.0 ± 7.5 **	17.5 ± 8.4 **
GTP (IU/L)	48.3 ± 30.5	17.9 ± 12.0 **	16.6 ± 7.4 **	16.3 ± 7.5 **	17.9 ± 8.2 **
Antidiabetic drugs use (%, n)	50.0(24/48)	6.2(3/48)	6.2(3/48)	12.5(6/48)	14.6(7/48)
Antihypertensive drugs use (%, n)	66.7(32/48)	14.6(7/48)	25.0(12/48)	27.1(13/48)	22.9(11/48)
Lipid-lowering drugs use (%, n)	29.2(14/48)	16.7(8/48)	16.7(8/48)	25.0(12/48)	27.1(13/48)

TBWL, total body weight loss; EBWL, excessive body weight loss; BMI, Body mass index; BW, body weight. * *p* < 0.05; ** *p* < 0.01 (indicate significant changes compared to pre-LSG assessed by ANOVA).

**Table 2 nutrients-16-03926-t002:** Time-course changes in body composition.

	Pre-LSG	6 Months	1 Year	2 Years	3 Years
FM (kg)	53.2 ± 15.1	30.0 ± 13.1 *	27.8 ± 13.0 *	29.2 ± 15.5 *	32.1 ± 15.6 *
FM (%)	47.3 ± 6.5	35.7 ± 10.0 *	33.9 ± 9.4 *	34.6 ± 10.5 *	37.2 ± 10.0 *
Total MM (kg)	55.2 ± 10.2	48.9 ± 9.4 *	48.6 ± 9.3 *	48.4 ± 9.8 *	48.1 ± 9.6 *
Total MM/BW	0.50 ± 0.06	0.60 ± 0.09 *	0.62 ± 0.09 *	0.61 ± 0.10 *	0.59 ± 0.09 *
Skeletal MM (kg)	32.4 ± 6.4	28.3 ± 5.9 *	28.1 ± 5.9 *	28.0 ± 6.2 *	27.9 ± 6.1 *
Skeletal MM/BW	0.29 ± 0.04	0.35 ± 0.06 *	0.36 ± 0.05 *	0.36 ± 0.06 *	0.35 ± 0.07 *
Upper Skeletal MM/BW	0.06 ± 0.01	0.07 ± 0.01 *	0.07 ± 0.01 *	0.07 ± 0.01 *	0.07 ± 0.01 *
Lower Skeletal MM/BW	0.16 ± 0.03	0.20 ± 0.03 *	0.20 ± 0.03 *	0.20 ± 0.03 *	0.20 ± 0.04 *
Body fluid	42.0 ± 9.6	37.2 ± 8.5 *	37.0 ± 8.5 *	36.8 ± 8.8 *	36.6 ± 8.7 *
Bone mineral content	2.95 ± 0.72	2.98 ± 0.57	2.96 ± 0.54	2.91 ± 0.54 *	2.86 ± 0.53 *
The ratio of extra cellular fluid	0.387 ± 0.009	0.394 ± 0.009 *	0.394 ± 0.010 *	0.393 ± 0.009 *	0.391 ± 0.008 *

FM, fat mass; MM, muscle mass. * *p* < 0.01 (indicates significant changes compared to pre-LSG assessed by ANOVA).

**Table 3 nutrients-16-03926-t003:** Multiple linear regression models with body weight as the dependent variable.

Variables	*t*-Value	*p*-Value
FM	5.44	<0.01 *
% FM	−0.40	0.69
Total MM	1.01	0.31
%Total MM	−0.29	0.77
Upper Skeletal MM/BW	0.91	0.37
Lower Skeletal MM/BW	−0.59	0.56

FM, fat mass; MM, muscle mass, * *p* < 0.01 significant correlation between factors.

## Data Availability

The original contributions presented in the study are included in the article, further inquiries can be directed to the corresponding authors.

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
