# Peer review of "Positive Changes in Body Composition and Profiles of Individuals with Diabetes 3 Years Following Laparoscopic Sleeve Gastrectomy in Japanese Patients with Obesity"

_nutrients, 2024, doi:10.3390/nu16223926_

Round 1
Reviewer 1 Report
Comments and Suggestions for Authors
Commentary
Thanks to the Editors for the opportunity to review this paper. In this manuscript, the authors present their retrospective study to evaluate the changes on weight, glucose metabolism, and body composition over a 3-year period in obese Japanese patients following laparoscopic sleeve gastrectomy. There are limited studies indicating long-term changes in body composition, including body fat and muscle mass, therefore the topic is hot and important.
Major issues:
- In a period of 8 years of study the authors enrolled 102 patients, but only 48 were included in the current study. As they reported, this is a great limit of the study. Please explain better with a flowchart the selection of the patients.
- Materials and Methods 2.1: in this paragraph there are data that should be included in the section Results 3.1, as information about gender, age, BMI, number of patients affected by diabetes and arterial hypertension. Medications for diabetes and hypertension are not important for the aim of the study and should be eliminated. This paragraph should include the methods about reporting weight loss: change in BMI (ΔBMI): ΔBMI = (Initial BMI) – (Postop BMI), percent of total weight loss (%TWL): %TWL = [(Initial Weight) – (Postop Weight)] / [(Initial Weight)] x 100, percent excess weight loss (%EWL): %EWL = [(Initial Weight) – (Postop Weight)] / [(Initial Weight) – (Ideal Weight)] as described in is described in ASMBS (Brethauer SA, et al; ASMBS Clinical Issues Committee. Standardized outcomes reporting in metabolic and bariatric surgery. Surg Obes Relat Dis. 2015 May-Jun;11(3):489-506. doi: 10.1016/j.soard.2015.02.003.). So please edit the manuscript according to the guidelines.
- Materials and Methods 2.2: please provide more details on the operative procedures.
- Materials and Methods 2.3: please provide information about the time of followup in which blood samples were collected.
- Results 3.1: table 1 and table 2 should be joint in only one table reporting data about bioelectrical impedance analysis (MM, FM….) pre and postoperative. Please report data about the difference in men and women and the p value.
- Results 3.3: as suggested above, results on FM and %FM should be reported in the table and discussed in this paragraph.
- Results 3.4: abbreviations in this paragraph have not been reported in materials and methods.
- Results 3.5: this paragraph could be joined to results 3.4 paragraph.
- Results 3.7: please provide a table or a graph about the multiple regression analysis.
- Discussion line 216: is not allowed to report unpublished data, please provide appropriate literature reference.
- Discussion line 219: the sentence “MM could be associated with a higher risk of developing diabetes” is unclear and not supported by the following discussion. Please edit this sentence.
- Discussion line 262: please provide the abbreviations for VAT and SAT.
- Please extend the section Discussion report a comparison with articles about changes after bariatric surgery in non Japanese subjects.

Author Response
Thank you very much for your comments. We present point-by-point responses to your questions and comments.
Comments 1: In a period of 8 years of study the authors enrolled 102 patients, but only 48 were included in the current study. As they reported, this is a great limit of the study. Please explain better with a flowchart the selection of the patients.
Response 1: Thank you very much for the comments. Regarding the high interruption rate in the present study, it is due to the current situation of bariatric surgery in Japan and COVID-19 pandemic between 2019 and 2022. There are few hospitals in Japan that perform bariatric surgery, and in fact, many of our hospital's patients come from outside the prefecture. As a result, about one-third of patients who come from far away transfer to a nearby hospital within a few years after bariatric surgery. In addition, there were several interruptions of medical treatment due to COVID-19 pandemic between 2019 and 2022. In the present study, a total of 95 consecutive patients who performed LSG were initially recruited. There were 47 patients who lost follow-up due to the following reasons: transfer to another hospital (n =23), COVID-19 infection (n =5), and treatment interruption (n=19). 48 patients (18 male and 30 female) ultimately participated in the study. In the revised MS, we have described the points with a flowchart of the study participants in Materials and Methods and Figure (Materials and Methods: Page 2, Line 85) (Figure 1).
Comments 2: Materials and Methods 2.1: in this paragraph there are data that should be included in the section as information about gender, age, BMI, number of patients affected by diabetes and arterial hypertension. Medications for diabetes and hypertension are not important for the aim of the study and should be eliminated.
Response 2: We appreciate the comments. 48 patients (19 male and 29 females, mean body mass index [BMI] = 43.7±8.7 kg /m2, mean age = 42.2±9.3 years) ultimately participated in the study. Among 48 patients, 24 patients have T2DM, and 32 patients have hypertension. We have presented the data about gender, age, BMI, number of patients affected by diabetes and arterial hypertension in this paragraph (Materials and Methods: Page 2, Line 76). In addition, we have deleted the description of medications for diabetes and hypertension.
Comments 3: This paragraph should include the methods about reporting weight loss: change in BMI (ΔBMI): ΔBMI = (Initial BMI) – (Postop BMI), percent of total weight loss (%TWL): %TWL = [(Initial Weight) – (Postop Weight)] / [(Initial Weight)] x 100, percent excess weight loss (%EWL): %EWL = [(Initial Weight) – (Postop Weight)] / [(Initial Weight) – (Ideal Weight)] as described in ASMBS (Brethauer SA, et al; ASMBS Clinical Issues Committee. Standardized outcomes reporting in metabolic and bariatric surgery. Surg Obes Relat Dis. 2015 May-Jun;11(3):489-506. doi: 10.1016/j.soard.2015.02.003.). So please edit the manuscript according to the guidelines.
Response 3: We sincerely appreciate the comments. We have added a description the methods about weight loss: change in BMI (ΔBMI), %TWL and %EWL according to previous report (Surg Obes Relat Dis. 2015; 11:489-506) (Materials and Methods: Page 3, Line 115).
Comments 4: Materials and Methods 2.2: please provide more details on the operative procedures.
Response 4: We agree with the comments. The surgical techniques for the LSG procedures have been previously reported. In summary, following visual access to the intraperitoneal cavity using a 10mm Visiport trocar (US Surgical, Norwalk, USA) placed 18cm below the xiphoid process, the abdomen was inflated to 15mmHg. Subsequently, a 15mm port, a 12mm port, and two 5mm ports were positioned in the upper abdomen, and a Nathanson liver retractor (Automated Medical Products Corp, Edison, USA) was introduced through a 5 mm skin incision in the subxiphoid area to hold back the left lateral segment of the liver. The vessel sealing system (LigaSure system, Valleylab, Boulder, USA) was used to dissect the greater omentum from 5 cm proximal to the pyloric ring to the angle of His. Following the placement of a 10.5mm (32-Fr) endoscope with or without a 15mm (45-Fr) endoscopy tube directly along the lesser curvature of the stomach, endoscopic linear staplers (45 or 60mm in length, Endogean, US Surgical) were applied in a sequential manner, commencing 6cm from the pylorus. Subsequently, the staple lines were reinforced to avoid bleeding and leakage. The specimen was obtained via the 15mm port site. Ultimately, oral endoscopy was used to examine stenosis of the remnant stomach as well as hemorrhaging and air leakage from the staple line. We have described the point in the Materials and Methods (Materials and Methods: Page 3, Line 89).
Comments 5: Materials and Methods 2.3: please provide information about the time of follow up in which blood samples were collected.
Response 5: We sincerely appreciate the comments. In the present study, blood samples were collected prior to LSG at the first consultation, 6month, 1 year, 2 years, and 3 years after LSG. We have described the point in the Materials and Methods (Materials and Methods: Page 3, Line 111).
Comments 6: Results 3.1: table 1 and table 2 should be joint in only one table reporting data about bioelectrical impedance analysis (MM, FM….) pre and postoperative. Please report data about the difference in men and women and the p value.
Response 6: Thank you very much for the comments. In the revised MS, we have combined Table 1 and table 2 in one table. In addition, there is no difference between men and women regarding changes of BW, BMI, FM and MM in the present study. We have presented the data the difference in men and women and the p value in Results 3.1, 3.3 and 3.4. (Results: Page 4 Line 149, Page 4, Line 172 and Page 5, Line 186).
Comments 7: Results 3.3: as suggested above, results on FM and %FM should be reported in the table and discussed in this paragraph.
Response 7: Thank you very much for the comments. We have presented the results on FM and %FM as Table 2 and described them in this paragraph (Results: Page 5, Line 174) (Table 2).
Comments 8: Results 3.4: abbreviations in this paragraph have not been reported in materials and methods.
Response 8: We sincerely appreciate the comments. We have reported abbreviations in materials and methods.
Comments 9: Results 3.5: this paragraph could be joined to results 3.4 paragraph.
Response 9: We agree with the comments. We have combined Results 3.4 and Results 3.5 in the revised MS (Results: Page 5, Line 176).
Comments 10: please provide a table or a graph about the multiple regression analysis.
Response 10: Thank you very much for the comments. We have presented the data of multiple regression analysis as Table.3. (Results: Page 6, Line 224) (Table 3).
Comments 11: Discussion line 216: is not allowed to report unpublished data, please provide appropriate literature reference.
Response 11: We appreciate the comments. We have deleted the reference and statement (Discussion: Page 7, Line 247).
Comments 12: Discussion line 219: the sentence “MM could be associated with a higher risk of developing diabetes” is unclear and not supported by the following discussion. Please edit this sentence.
Response 12: We agree with the comments. The loss of MM could be associated with a higher risk of developing diabetes. We have appropriately edited this sentence (Discussion: Page 7, Line 248).
Comments 13: Discussion line 262: please provide the abbreviations for VAT and SAT.
Response 13: We sincerely appreciate the comments. VAT is an abbreviation for visceral adipose tissue and SAT means subcutaneous adipose tissue. We have provided abbreviations for VAT and SAT (Discussion: Page 8, Line 291).
Comments 14: Please extend the section Discussion report a comparison with articles about changes after bariatric surgery in non-Japanese subjects.
Response 14: Thank you very much for the comments. Loss of muscle mass and sarcopenia after LSG is described in non-Japanese subjects (Obes Surg. 2022; 32:3830-3838). We have discussed the point in the Discussion (Discussion: Page 8, Line 295).
Reviewer 2 Report
Comments and Suggestions for Authors
I greatly appreciated this article, for its clarity and concision. The topic is interesting, because lower cases of LSG have been performed in Japan in comparison to western countries, and limited studies report the long-term effects of LSG on body weight (BW), body composition (using Bioelectrical impedance analysis), lipid and diabetic profiles. Moreover, the duration of the follow-up is more than remarkable.
The article reports the data derived from the FU of 42 obese Japanese patients following laparoscopic sleeve gastrectomy (LSG). Body weight, diabetic and lipid parameters, fat mass (FM), muscle mass, and %FM have been tracked before surgery and 6 months, 1 year, 2 years, and 3 years after LSG. Significant improvements in fat mass and glycemic and lipid metabolism have been recorded over a 3-yrs period, while %muscle mass increased at 6 month and then remained stable, resulting inversely correlated with body weight and HbA1c. Moreover, all patients on antihypertensive or antidiabetic medications reduced or stopped the treatments during the observation period.
Abstract: I suggest mentioning also the improvements in blood lipids and blood pressure
Introduction: clearly exposes the background of the research
Materials and Methods: thoroughly described
Results: concise but properly exposed thanks to the tables and the graphics
Discussion: appropriately proposes the results and their supporting mechanisms
Limitations: authors recognize that the sample is small, and the study of a larger sample is warranted.
Author Response
Thank you very much for your comments. We present point-by-point responses to your questions and comments.
Comments 1: Abstract: I suggest mentioning also the improvements in blood lipids and blood pressure.
Response 1: We sincerely appreciate the comments. In the revised MS, we have added the description of the improvements in blood lipids and blood pressure in Abstract (Abstract Page 1, Line 22).
Comments 2:
Introduction: clearly exposes the background of the research
Materials and Methods: thoroughly described
Results: concise but properly exposed thanks to the tables and the graphics
Discussion: appropriately proposes the results and their supporting mechanisms
Limitations: authors recognize that the sample is small, and the study of a larger sample is warranted.
Response 2: Thank you very much for your comments.
Reviewer 3 Report
Comments and Suggestions for Authors
BP control improvement and lipid and carbohydrate disturbances are expected outcomes after bariatric surgery. The only novelty is the Japan cohort.
An interesting observation is the extensive loss of muscle mass in the upper extremities.
The analysis of outcomes is poor and has to be improved.
The abstract does not include the number of assessed subjects.
Material and Methods: It would be much better to show patients' characteristics in a table. It is unclear how many were lost to follow-up.
Data in Table 1 are duplicated in Table 2.
line 127 side effects or adverse events?
The specific time points did not show data concerning antihypertensive therapy, lipid-lowering therapy, and antidiabetic treatment.
How to analyze correlations when patients have modified therapies over time?
Study outcomes have to be better defined.
There is no analysis of weight regain.
The conclusion is too simple.
Author Response
Thank you very much for your comments. We present point-by-point responses to your questions and comments.
Comments 1: The analysis of outcomes is poor and has to be improved.
Response 1: We sincerely appreciate the comments. We have presented time-course change of body composition as Table 2 and described them in Results (Results: Page 5, Line 176) (Table.2). In addition, we have presented the data of multiple regression analysis as Table 3. (Results: Page 6, Line 224) (Table 3).
Comments 2: The abstract does not include the number of assessed subjects.
Response 2: Thank you very much for the comments. Body weight, diabetic parameters such as glycated hemoglobin (HbA1c), and electrical impedance analysis was used to assess body composition in forty-eight obese Japanese patients in the present study. We have included the number of assessed subjects in the abstract (Abstract: Page 1, Line 20).
Comments 3: Material and Methods: It would be much better to show patients' characteristics in a table. It is unclear how many were lost to follow-up.
Response 3: We sincerely appreciate the comments. We have described patients' characteristics in a table1 (Table 1). In addition, we have added a description of the lost follow-up patients with a flowchart as Figure 1 (Material and Methods: Page 2, Line 85) (Figure 1).
Comments 4: Data in Table 1 are duplicated in Table 2.
Response 4: Thank you very much for the comments. We have combined Table 1 and table 2 in one table as new Table 1 (Table 1).
Comments 5: line 127 side effects or adverse events?
Response 5: Thank you very much for the comments. We have appropriately revised the point (Results: Page 4, Line 146).
Comments 6: The specific time points did not show data concerning antihypertensive therapy, lipid-lowering therapy, and antidiabetic treatment.
Response 6: We appreciate the comments. The number of taking antihypertensive therapy, lipid-lowering therapy, and antidiabetic treatment are 3, 12, and 8 at 6month-2 years after LSG for each. 3 years after LSG, the number of antihypertensive therapies, lipid-lowering therapy, and antidiabetic treatment are 7, 11, and 13 for each. However, the reviewer 1 indicated that medications for diabetes and hypertension are not important for the aim of the study and should be eliminated. Thus, we have not presented the point in the revised MS.
Comments 7: How to analyze correlations when patients have modified therapies over time?
Response 7: Thank you very much for the comments. It is an important point, and we have discussed the influence of modified therapies on body composition (Limitation: Page 8, Line 312).
Comments 8: Study outcomes have to be better defined.
Response 8: Thank you very much for the comments. We have appropriately defined the point throughout the manuscript.
Comments 9: There is no analysis of weight regain.
Response 9: Thank you very much for the comments. It is an important point. In the present study, the significant weight regain couldn't be observed in 3 years as Table 1 and Table 2. We would like to analyze the weight regain and body composition longer in the future.
Comments 10: The conclusion is too simple.
Response 10: Thank you very much for the comments. We have described the conclusion in greater detail in the revised manuscript (Discussion: Page 7, Line 300).
Round 2
Reviewer 3 Report
Comments and Suggestions for Authors
Please look at the paper title including diabetic profiles. Therefore, please add at least percentages of subjects on antidiabetic treatment for time points.
The data should be presented as the outcome, and described in the result section.
Author Response
Comment 1: Please look at the paper title including diabetic profiles. Therefore, please add at least percentages of subjects on antidiabetic treatment for time points.
The data should be presented as the outcome and described in the result section.
Response 1: Thank you very much for your comments. We sincerely appreciate and agree with the comment. The percentages of taking antidiabetic treatment are 50%, 6%, 13%, and 15% for pre-surgery, 1 year, 2 years and 3 years after LSG treatment. We have added the points in the result section (Results: Page 4, Line 164).
Round 3
Reviewer 3 Report
Comments and Suggestions for Authors
The paper has been improved, however, I cannot accept the lack of anti-diabetic medication in the specific time points looking at the title. Please add them (percentages on or without antidiabetic medications).
Similar for antihypertensive and lipid-lowering therapies.
They should be reflected in the result section.
Author Response
Comment 1: The paper has been improved, however, I cannot accept the lack of anti-diabetic medication in the specific time points looking at the title. Please add them (percentages on or without antidiabetic medications). Similar for antihypertensive and lipid-lowering therapies. They should be reflected in the result section.
Response: Thank you very much for your comments. In the present study, 50% (24/48) patients were taking anti-diabetic medications (DPP-4 inhibitors, 12 patients; glucagon-like peptide 1 receptor agonists, 5 patients; SGLT2 inhibitors, 6 patients; metformin, 13 patients; sulfonylureas, 5 patients; insulin, 3 patients; and others, 7 patients) as type-2 diabetes prior to bariatric surgery. In addition, 67% (32/48) patients were taking antihypertensive medications (angiotensin II receptor blockers, 18 patients; mineralocorticoid receptor antagonists, 1 patient; calcium channel blockers, 16 patients; and others, 7 patients). 29% (14/48) patients were taking lipid-lowering medications (Statins, 11 patients; Fibrates, 3 patients; and others, 1 patient).
As shown Table 1, the proportion of patients who were taking any glucose-lowering, anti-hypertension, lipid-lowering medications all decreased 1, 2, and 3 years after surgery compared to the levels at pre-surgery, respectively (Table 1). The percentage of patients with diabetes remission was 62% (16/26) at 3 years after surgery. In addition, the percentage of patients to normal blood pressure without antihypertensive drugs was 56% (18/32).
In the revised MS, we have described the points in the result section and Table.1 (Results: Page 4, Line 166-, Table 1).